META-RESEARCH ARTICLE

# Epidemiology and reporting characteristics of preclinical systematic reviews

**Victoria T. Hunniford**[1], **Joshua Montroy**[1], **Dean A. Fergusson**[1,2], **Marc T. Avey**[3], **Kimberley E. Wever**[4], **Sarah K. McCann**[5], **Madison Foster**[1], **Grace Fox**[1], **Mackenzie Lafreniere**[1], **Mira Ghaly**[1], **Sydney Mannell**[1], **Karolina Godwinska**[1], **Avonae Gentles**[1], **Shehab Selim**[2], **Jenna MacNeil**[2], **Lindsey Sikora**[6], **Emily S. Sena**[7], **Matthew J. Page**[8], **Malcolm Macleod**[7], **David Moher**[9], **Manoj M. Lalu**[1,10]*

1 Clinical Epidemiology Program, Blueprint Translational Research Group, Ottawa Hospital Research Institute, Ottawa, Ontario, Canada, 2 Department of Medicine, University of Ottawa, Ottawa, Ontario, Canada, 3 ICF International Incorporated, Ottawa, Canada, 4 SYstematic Review Center for Laboratory animal Experimentation (SYRCLE), Radboud Institute for Health Sciences, Radboud University Medical Center, Nijmegen, the Netherlands, 5 QUEST Center for Transforming Biomedical Research, Berlin Institute of Health (BIH) and Charité—Universitätsmedizin Berlin, Berlin, Germany, 6 Health Sciences Library, University of Ottawa, Ottawa, Canada, 7 Centre for Clinical Brain Sciences, University of Edinburgh, Edinburgh, United Kingdom, 8 School of Public Health and Preventive Medicine, Monash University, Melbourne, Australia, 9 Centre for Journalology, Clinical Epidemiology Program, Ottawa Hospital Research Institute, Ottawa, Canada, 10 Department of Anesthesiology and Pain Medicine, The Ottawa Hospital, University of Ottawa, Ottawa, Ontario, Canada

* mlalu@toh.ca

**Data Availability Statement:** All relevant data files are either within the Supporting information or available from the following links: the DOI to the

## Abstract

In an effort to better utilize published evidence obtained from animal experiments, systematic reviews of preclinical studies are increasingly more common—along with the methods and tools to appraise them (e.g., SYstematic Review Center for Laboratory animal Experimentation [SYRCLE's] risk of bias tool). We performed a cross-sectional study of a sample of recent preclinical systematic reviews (2015–2018) and examined a range of epidemiological characteristics and used a 46-item checklist to assess reporting details. We identified 442 reviews published across 43 countries in 23 different disease domains that used 26 animal species. Reporting of key details to ensure transparency and reproducibility was inconsistent across reviews and within article sections. Items were most completely reported in the title, introduction, and results sections of the reviews, while least reported in the methods and discussion sections. Less than half of reviews reported that a risk of bias assessment for internal and external validity was undertaken, and none reported methods for evaluating construct validity. Our results demonstrate that a considerable number of preclinical systematic reviews investigating diverse topics have been conducted; however, their quality of reporting is inconsistent. Our study provides the justification and evidence to inform the development of guidelines for conducting and reporting preclinical systematic reviews.

## Introduction

Systematic reviews and meta-analyses are essential tools for synthesizing evidence in a transparent and reproducible manner [1]. They provide a rigorous method to comprehensively

OSF project containing this data: DOI 10.17605/OSF.IO/W3JUH and the OSF link: https://osf.io/tkcvs/.

**Funding:** The authors received no specific funding for this work.

**Competing interests:** I have read the journal's policy and the authors of this manuscript have the following competing interests: Dr. Malcolm Macleod received a grant from the NC3Rs as salary support; Dr. Manoj Lalu reports support from The Ottawa Hospital Anesthesia Alternate Funds Association and from a University of Ottawa Junior Research Chair in Innovative Translational Research during the conduct of the study; and Dr. Emily Sena reports support from Stroke Association (SAL-SNC 18\1003) during the conduct of the study.

**Abbreviations:** CAMARADES, Collaborative Approach to Meta-Analysis and Review of Animal Data from Experimental Studies; DistillerSR, Distiller Systematic Review Software; MOOSE, Meta-analyses Of Observational Studies in Epidemiology; PICO, population, intervention, comparison, outcome; PRISMA, Preferred Reporting Items for Systematic reviews and Meta-Analyses; SYRCLE, SYstematic Review Center for Laboratory animal Experimentation.

identify, summarize, evaluate, and appraise the available evidence on a topic. Clinical systematic reviews have been used for over 3 decades by policy makers, clinicians, and other stakeholders to inform decision-making and evidence-based practice [2]. More recently, systematic reviews of preclinical research have been used to identify, summarize, evaluate, and appraise laboratory-based studies [3–5]. With an emerging recognition of the importance of rigor and reproducibility, preclinical systematic reviews have been recognized to provide important information to inform the translational pathway of novel therapeutics [6].

The popularity of preclinical systematic reviews has been growing over the past decade. Groups such as the Collaborative Approach to Meta-Analysis and Review of Animal Data from Experimental Studies (CAMARADES, http://www.dcn.ed.ac.uk/camarades/) and SYstematic Review Center for Laboratory animal Experimentation (SYRCLE, https://www.syrcle.nl/) have been established in part to provide support for researchers conducting systematic reviews and meta-analyses of experimental animal studies. Although the Preferred Reporting Items for Systematic reviews and Meta-Analyses (PRISMA) guidelines have been critical to the proper reporting of systematic reviews [7,8], they lack specificity for reporting systematic reviews of preclinical research. There are fundamental differences between preclinical and clinical systematic reviews of interventions. For instance, subjects are laboratory animals instead of humans, and studies employ different procedures compared with clinical studies (e.g., inducing the disease in animals to mimic the human condition, humanely killing animals to evaluate outcomes, etc.). Therefore, species- and strain-specific effects, along with unique elements of data abstraction, may need to be considered. In addition, preclinical systematic reviews are sometimes performed as precursors to attempted clinical translation of a novel therapy. Therefore, in addition to risk of bias, exploring construct validity of included primary studies (i.e., how well the animal models mimic the disease of interest) may provide important insights on the potential for translation [9]. Finally, it is commonplace to include primary studies with multiple experiments, each with many experimental arms; this requires both significant efforts to identify specific data to include, as well as special considerations for how data will be handled when analyzed. As this field continues to develop, it is important to audit practices and ensure that norms of systematic review conduct and reporting are adhered to. Several reports have been performed to identify and characterize published systematic reviews of preclinical experiments. The most recent evaluation published by Mueller and colleagues [5] summarized 512 preclinical systematic reviews published between 1989 and 2013. They found the quality of reporting in these reviews low, the majority not reporting assessing the risk of bias or heterogeneity of included studies.

The prevalence and state of reporting of preclinical systematic reviews has not been formally evaluated since 2014 (by Mueller and colleagues) [5]. It is unknown if the reporting of these reviews has improved. Thus, a contemporary assessment of published preclinical systematic reviews is warranted. This assessment will help inform the future development of reporting guidelines designed specifically for preclinical systematic reviews, as an extension to the PRISMA statement.

## Methods

The protocol for this study was posted on Open Science Framework (https://osf.io/9mzsv/) and is part of a larger program of research to generate an extension for the PRISMA guidelines specific to preclinical *in vivo* animal experiments (https://osf.io/kv3ed/). It is important to note that the reporting assessment described here is not an extension to PRISMA; instead, the results of this study will be used to inform the development of the aforementioned preclinical extension (Delphi protocol: https://osf.io/4wy3s/).

## Eligibility criteria

All preclinical systematic reviews that investigated interventions using *in vivo* animal research were eligible. Systematic reviews with or without meta-analyses were included. We only considered *in vivo* animal research, as we intend to use the findings of this work to inform guidelines that are specifically for reporting preclinical systematic reviews of *in vivo* experiments.

We included systematic reviews that met at least 3 of the 4 following statements according to the 2009 PRISMA statement [9]: (a) a clearly stated set of objectives with explicit methodology; (b) a systematic search was employed; (c) an assessment of validity of findings (e.g., risk of bias assessment) was conducted; and (d) systematic presentation, synthesis of characteristics and findings of the included studies.

We defined preclinical as research investigating medically relevant interventions that is conducted using nonhuman models, with an intention to progressing to testing efficacy in human participants prior to being approved. Models were limited to *in vivo* (in living animals) experiments; thus, reviews of solely *in vitro* (in cells, microorganisms, or biological molecules) or *ex vivo* (in tissue removed from a living subject) experiments were excluded. We did not limit inclusion by the domain of the preclinical study or potential clinical scope. We included any potentially therapeutic intervention including, but not limited to substances, antibodies, vaccines, gene therapies, technical devices and other non-pharmacologic interventions (e.g., surgical procedures or dietary interventions), combination therapies, or novel enhancements of already established clinical therapies. We did not limit inclusion based on comparator groups or outcomes measured. We excluded clinical systematic reviews, meta-analyses of datasets (e.g., large databases) not gathered from literature or retrieved from non-laboratory studies, as well as non-English articles, scoping reviews, narrative reviews, rapid reviews, and systematic reviews published only as conference abstracts/proceedings.

## Information sources and search strategy

A comprehensive literature search was developed and conducted in conjunction with an information specialist. We searched MEDLINE via Ovid, Embase via Ovid, and Toxline for preclinical systematic reviews of *in vivo* animal research (2015 to 2018 inclusive; searches performed on January 13, 2017 and updated January 24, 2018 and March 21, 2019). The details of the search strategy are available in S1 Appendix.

## Screening and data extraction

The literature search results were uploaded to Distiller Systematic Review Software (DistillerSR, Evidence Partners, Ottawa, Canada). DistillerSR is cloud-based software that manages references and provides customized reports for accurate review. Titles, abstracts, and full text were screened for inclusion by 2 independent reviewers using the eligibility criteria outlined above. Disagreements were resolved through consensus or by a third party, if necessary. Where titles and abstracts appeared to meet the inclusion criteria or where there was uncertainty, we reviewed the full text. Prior to the formal full-text screening, a calibration test of 13 systematic reviews was performed to refine the screening form to ensure no misinterpretation of the eligibility criteria. Inter-rater agreement was assessed (Cohen's kappa coefficient). The reasons for article exclusion at the full-text level were recorded. The study selection process was documented using the 2009 PRISMA flow diagram.

After identifying all eligible preclinical systematic reviews, we extracted data in duplicate with conflicts resolved by consensus or a third party. Prior to the formal data extraction, a pilot test on 13 reviews was performed to refine the data form and to ensure a high level of

inter-rate agreement. The extracted study characteristics included details about the publication (corresponding author's name, their contact information, the country their institution was located in, and publication year), the animal species investigated, and the disease domain being investigated (e.g., cardiovascular disease). We extracted the number of *in vivo* publications included in each review, the sources of funding, the category (pharmacological or non-pharmacological), and specific name of interventions.

## Assessment of reporting

We next evaluated the quality of reporting in a random sample of 25% included systematic reviews. This sample size was chosen based on available resources. These were selected using the embedded randomize function in DistillerSR. Our aim was to assess the reporting in preclinical systematic reviews; thus, we only selected studies in which the majority of data were derived from preclinical *in vivo* studies. Two reviewers determined eligibility of the randomly selected studies—resampling ineligible reviews until the full sample of 25% was reached. Two reviewers then independently assessed the reporting in the sample of preclinical systematic reviews using a reporting checklist developed *a priori* (described below). The reporting assessment was piloted on 7 reviews from the sample to ensure that reviewers were evaluating the reporting items consistently. Any disagreements were resolved by discussion or by a third party when necessary.

## Generating checklist

To create the checklist for the reporting assessment, we consulted PRISMA 2009 [9], along with 4 sets of expert guidance in preclinical systematic review [3,10–12]. All items from each source were included in an omnibus draft checklist, where each item was framed as 1 or more "binary" items (that could be answered with a clear yes or no rather than having multiple questions or conditions). This list was compared against additional sources, PRISMA 2020 (preprint) [13] and several previously published assessments of reports of preclinical systematic reviews [4,5,14], to generate a list of items. Items from the list that appeared only relevant to clinical systematic reviews were further evaluated by our team. If these items could not be modified for the preclinical context, they were removed. After discussion and feedback from experts in the fields of systematic review reporting and preclinical research, a final checklist containing 46 items (51 with included subitems) was generated. This was a collaborative and iterative process including all coauthors over the course of several virtual meetings. It is important to note that this checklist is not intended to be an extension for PRISMA, but rather a list of key reporting items our team wanted to assess.

Items were arranged by the following manuscript sections: title, introduction, methods, results, discussion, and other. We did not evaluate items that are specific to review abstracts, as the guideline for abstracts vary substantially by journal. Items were assessed in each review as being reported ("yes") or not ("no"), or not applicable to the review ("NA"). If a review did not contain quantitative data (i.e., no meta-analysis), "NA" was selected for all items relating to quantitative data/analysis (e.g., report methods for extracting numerical data from report). One exception was an assessment of the review's main objective/question (which would ideally be presented in a population, intervention, comparison, outcome [PICO] format). This could be scored as a "yes," "no," or "partial," where "partial" represented some, but not all, relevant PICO items of the question being stated. This checklist was piloted and refined by 2 independent reviewers to improve its utility in practical application before using it to assess systematic review reporting. The checklist can be found in S1 Checklist.

## Data analysis

The collected data are presented using descriptive statistics (total counts, medians, and ranges), as well as narratively when appropriate.

## Results

### Search results

Our searches identified a total of 2,356 records (2015 to 2018, inclusive, Fig 1). We excluded 1,585 records on the basis of abstract screening. Of 771 full-text articles retrieved for further evaluation, 329 were excluded (40% being commentaries, editorials, or short review/reports). Of note, 73 articles included in the abstract level for having "systematic review" in the title were later excluded at the full-text stage as they were not systematic reviews. The agreement between reviewers across the screening stage was very high (kappa = 0.93). A total of 442 preclinical systematic reviews met our inclusion criteria.

### Descriptive characteristics of included systematic reviews

Twenty-seven percent of reviews were published in 2015, 15% in 2016, 20% in 2017, and 38% were published in 2018. Corresponding authors resided in 43 different countries (Fig 2, S1 Table). Three hundred and thirty-four reported on funding, with 248 having receiving funding from 1 or more sources; 86 reviews stated they had no funding (Table 1).

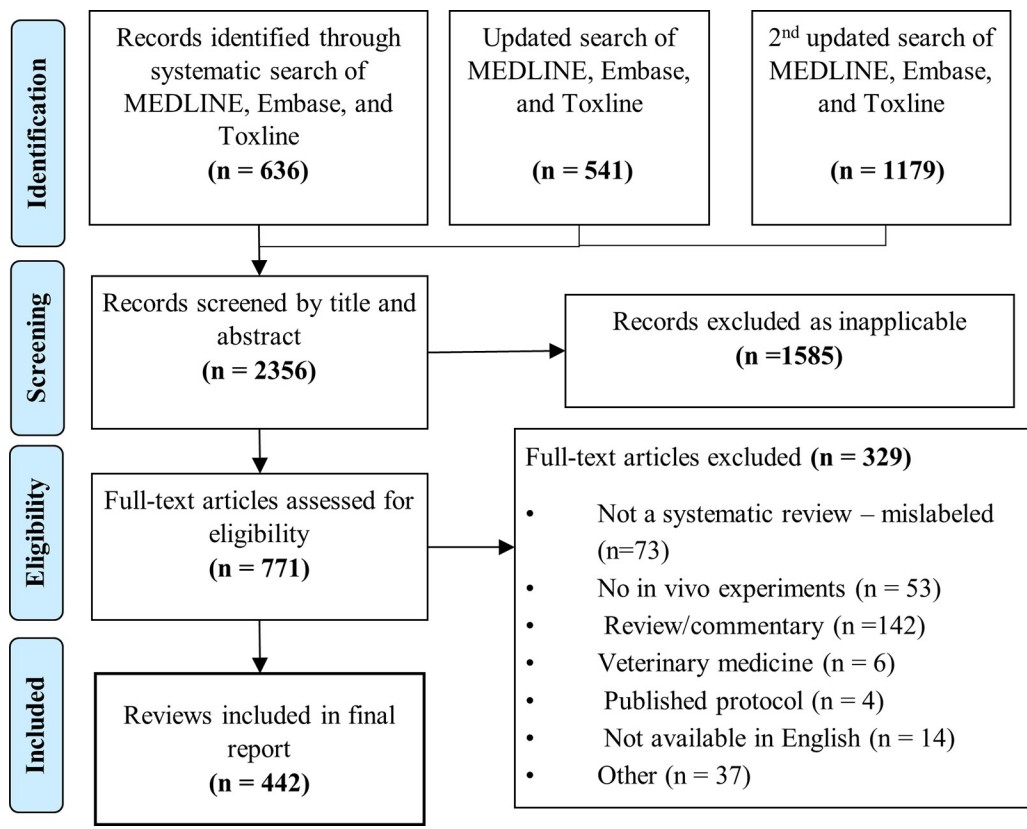

**Fig 1. PRISMA flow diagram of study selection process.** PRISMA, Preferred Reporting Items for Systematic reviews and Meta-Analyses.

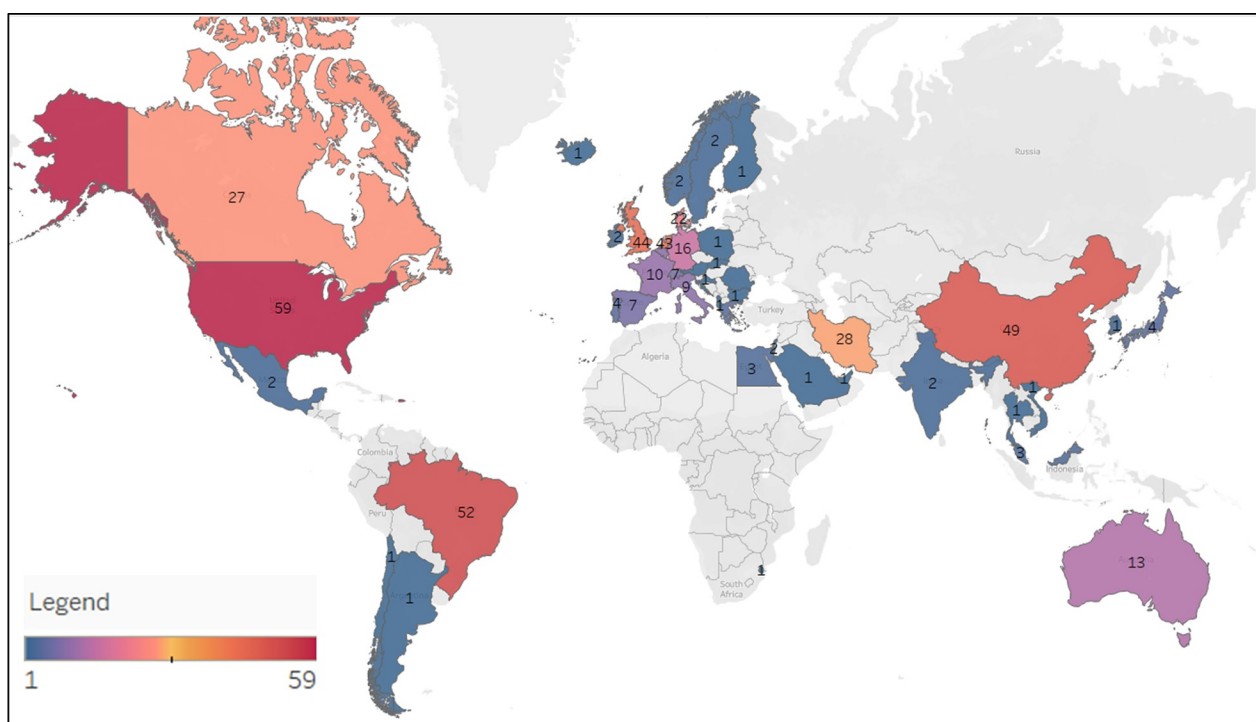

**Fig 2. Heatmap of the number of preclinical systematic reviews published by country.** No systematic reviews were published in the counties with gray coloring. The map was created using Tableau software.

Table 1. Descriptive characteristics of included preclinical systematic reviews.

| Category | Characteristic | Number (%), *n* = 442 |
|---|---|---|
| Year of publication | 2015 | 118 (27) |
| | 2016 | 66 (15) |
| | 2017 | 90 (20) |
| | 2018 | 168 (38) |
| Source of funding | Government | 163 (37) |
| | Academia | 78 (18) |
| | Foundation/charity | 73 (17) |
| | Pharmaceutical company | 13 (3) |
| | Hospital | 10 (2) |
| | Unfunded | 86 (20) |
| | Not reported | 108 (24) |
| Number of funding sources* | 1 | 174 (70) |
| | 2 | 55 (22) |
| | 3 | 15 (6) |
| | 4 | 4 (2) |
| Number of included publications | <10 | 87 (20) |
| | 10–100 | 318 (72) |
| | 100–300 | 28 (6) |
| | >300 | 4 (1) |
| | Not reported | 5 (1) |

*Percent calculation out of the number of funded reviews that reported the source (*n* = 248).

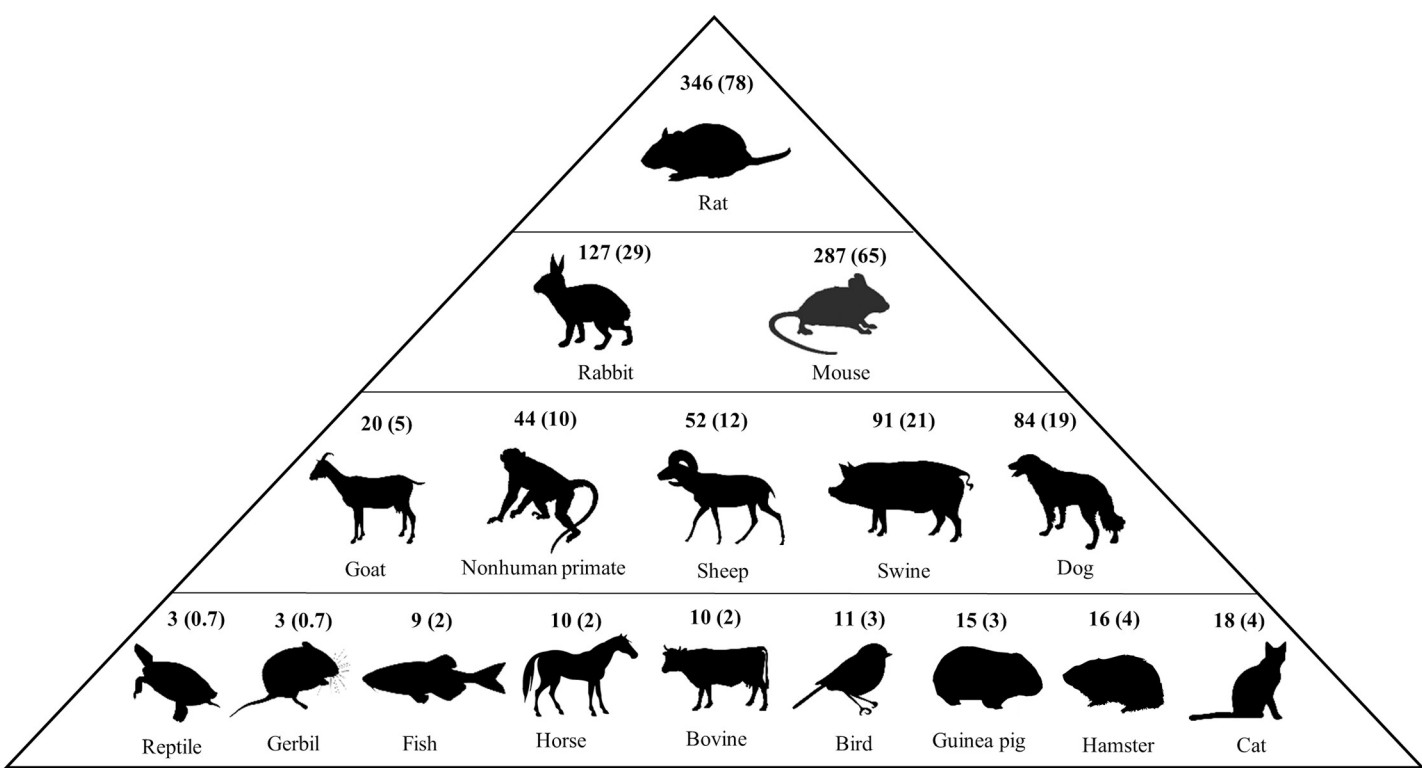

**Fig 3. Infographic of animal species/class/family included in the preclinical systematic reviews.** The structure of the pyramid and positioning of each species/class/family reflects their frequency and does not reflect hierarchy. [Values represent frequency and (%).]. The figure was created using non-copyrighted biological silhouettes retrieved from Phylopic.org.

The median number of primary publications included in the preclinical systematic reviews was 24 (range: 1 to 1,342). Eighty-two (19%) reviews contained data from both human studies (clinical trials and case studies) and *in vivo* animal experiments, and 97 (22%) reviews contained data from both *in vitro* and *in vivo* preclinical experiments. There were 26 animal species used in the included primary studies within the 442 preclinical systematic reviews (Fig 3, S2 Table). Twenty-three reviews did not report the species of animals used in their included primary studies. The median number of included species within the reviews was 2 (range: 1 to 9).

The preclinical systematic reviews investigated 23 different disease domains (Table 2). The most common were musculoskeletal system and connective tissue disorders (74 reviews; 17%), followed by disorders and afflictions to the nervous system (70 reviews; 16%). For the majority of reviews (363 reviews; 82%) the focus was on one disease domain, while 79 (18%) covered multiple disease domains.

Two-hundred and thirty-nine (54%) reviews reported pharmacological interventions, and 203 (46%) reported non-pharmacological interventions. Pharmacological interventions included substances like synthetic drugs, vaccines, and organic molecules. Within the 203 reviews that had a non-pharmacological intervention, 46 (10%) were cell therapies, 44 (10%) were surgery or invasive procedures, and 21 (5%) were medical physics interventions (e.g., ultrasound therapies) (Table 3).

## Reporting characteristics from a sample of systematic reviews

To assess the completeness of reporting within preclinical systematic reviews, we selected a random sample of 110 articles (25% of the 442 identified reviews): 64 of which evaluated

**Table 2. Disease domains investigated in the preclinical systematic reviews.**

| Category | Characteristic | Number (%), of n = 442 |
|---|---|---|
| Type of disease domain | Musculoskeletal system and connective tissue | 74 (17) |
| | Nervous system | 70 (16) |
| | Cardiovascular system | 66 (15) |
| | Endocrine, nutritional, and metabolic diseases | 42 (10) |
| | Cancer | 38 (9) |
| | Toxicology | 38 (9) |
| | Mental and behavior | 37 (8) |
| | Genitourinary system | 24 (5) |
| | Skin and subcutaneous tissue | 20 (5) |
| | Digestive system | 20 (5) |
| | Critical illness | 18 (4) |
| | Infectious and parasitic diseases | 17 (4) |
| | Respiratory system | 13 (3) |
| | Pain and analgesia | 13 (3) |
| | General and whole-body health | 10 (3) |
| | Conditions originating in the perinatal period | 9 (2) |
| | Pharmacokinetic, biological activity, and dose–response | 5 (1) |
| | Blood and immune disorders | 6 (1) |
| | Eye | 6 (1) |
| | Mouth | 6 (1) |
| | Congenital malformations | 5 (1) |
| | Surgery and imaging techniques | 4 (0.9) |
| | Auditory system | 1 (0.2) |
| Number of disease domains per review | 1 | 363 (82) |
| | 2 | 69 (16) |
| | 3 | 7 (2) |
| | >3 | 3 (0.7) |

**Table 3. Intervention and intervention subgroups evaluated in the preclinical systematic reviews.**

| Intervention Number (%), of n = 442 | Subgroup | Number (%), of n = 442 |
|---|---|---|
| Pharmacological 239 (54) | NA | |
| Non-pharmacological 203 (46) | Cell therapy | 46 (10) |
| | Surgery or invasive procedures | 44 (10) |
| | Medical physics | 21 (5) |
| | Dietary interventions | 13 (3) |
| | Blood transfusions or modifications | 11 (3) |
| | Animal model validation | 10 (2) |
| | Tactile stimulus interventions | 10 (2) |
| | Exercise and physical activity | 10 (2) |
| | Oxygen therapy | 7 (2) |
| | Gene therapy | 7 (2) |
| | Other | 24 (5) |

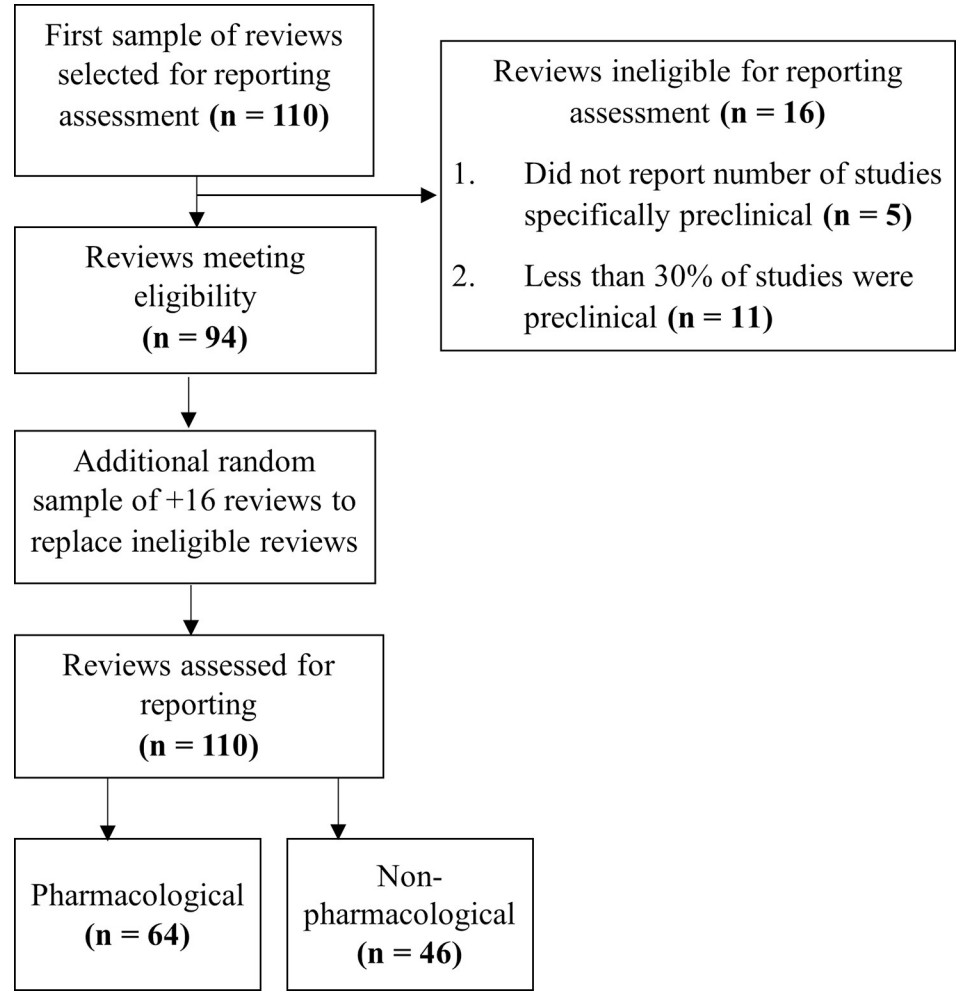

**Fig 4. Flowchart of reviews selected for the reporting assessment.**

pharmacological interventions and 46 evaluated non-pharmacological interventions (Fig 4). Inter-rater agreement between reviewers using our 46-item checklist (51 with subitems) was high (kappa = 0.89). The reporting assessments for the sample of 110 preclinical reviews is located in S3 Table.

## Reporting of title and introduction

Many (92; 84%) of the reviews indicated that the report was a "systematic review" in the title, while approximately half (54; 49%) reported that the review contained animal experiments in the title. Forty-five (41%) reviews reported both of these elements in the title. Within the introduction, most reviews described the human disease or health condition being modeled (104; 95%) and described the biological rationale for testing the intervention (106; 96%). Eighty (73%) reviews explicitly stated the review question(s) addressed (Fig 5).

## Reporting of methods

Twenty-two reviews (20%) reported a protocol had been developed *a priori*, of which 18 indicated where it could be accessed. Two of these reviews reported whether or not there were

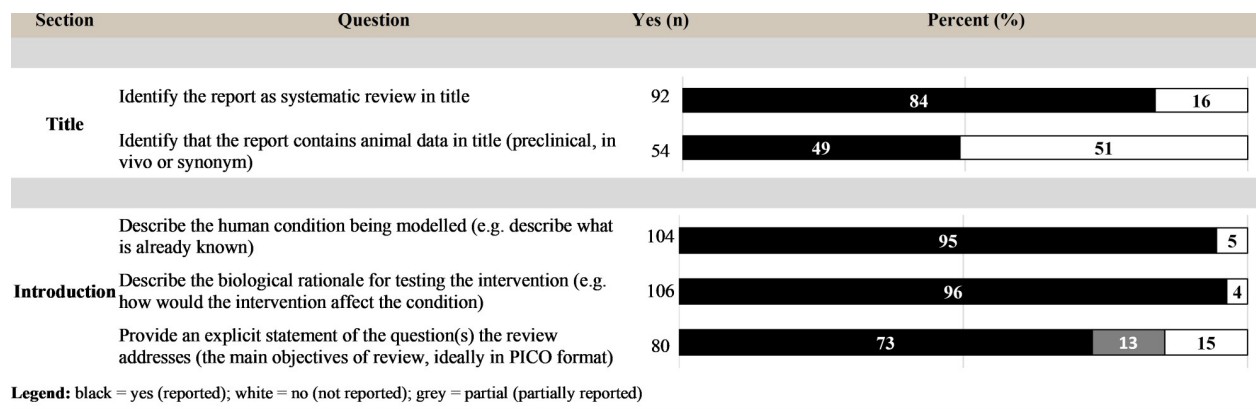

**Fig 5. Level of reporting (count and percent) for the items within the title and introduction sections.** PICO, population, intervention, comparison, outcome.

deviations in protocol. Of the reporting items dedicated to study inclusion criteria, the description of the eligible interventions/exposures was the highest reported item (93%). This was followed by reporting the animal species (65%), reporting outcomes (63%), and reporting the type/details of the animal models (53%) that could be included in the review. Thirty-five percent of reviews reported the eligible intervention timing (prevention versus rescue), although this item was not applicable to 31% of the reviews, as intervention timing was not a consideration. Sixty-nine percent reported the article inclusion limits (year of publication, type of article, and language restrictions).

In the methods section, 76% of reviews reported a full or a representative search strategy, and 72% described the study screening/selection process—of which 18 reviews (10%) reported the platform used to screen. Roughly two-thirds (62%) of reviews stated the number of independent screeners, while less than half (44%) reported the number of reviewers extracting data. Half of the reviews (49%) reported the methods and tool to measure study quality/risk of bias, while no (0%) reviews described methods for assessing construct validity (i.e., potential relevance to human health) [15]. For those in which it was applicable, 23% reported the methods for a publication bias assessment (Fig 6).

## Reporting of results

In the results section, almost all (106; 96%) the sampled reviews reported the number of included studies/publications, and 44% reported the number of independent experiments included in the analysis. The majority of reviews (86%) included a study selection flow diagram of the study selection process, and details such as study characteristics, animal species, and animal models, were generally well reported. For quality assessment measures, less than half (46%) reported the results of a risk of bias assessment, and 25% reported the results of assessing publication bias or that this assessment was not possible/done (Fig 7).

## Reporting of discussion

Within the discussion section, a minority of reviews (31%) discussed the impact of the risk of bias of the primary studies. Sixty-five percent of reviews discussed the limitations of the primary studies and outcomes to be drawn, while 56% of reviews discussed the limitations of the review itself. Twenty-one (19%) reviews reported on data sharing (Fig 8).

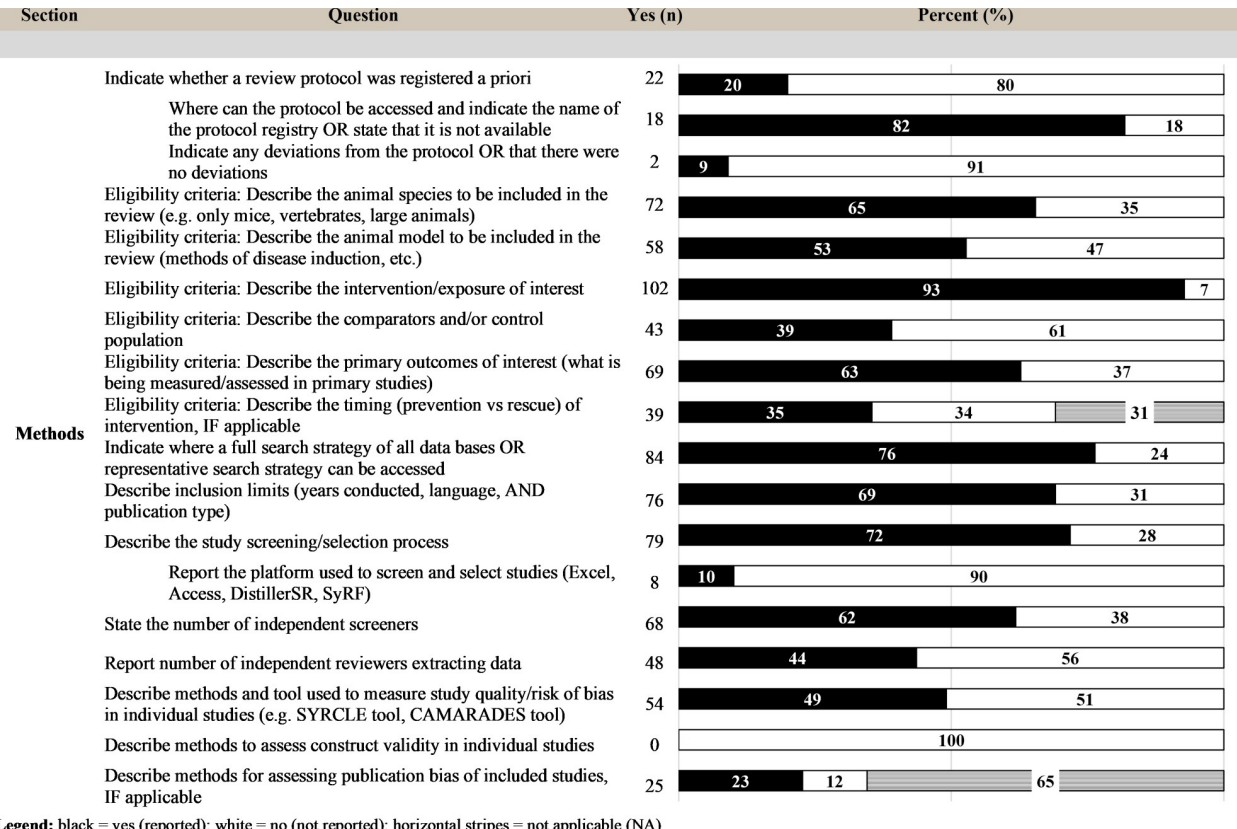

**Fig 6. Level of reporting (count and percent) for the items within the methods section.** CAMARADES, Collaborative Approach to Meta-Analysis and Review of Animal Data from Experimental Studies; DistillerSR, Distiller Systematic Review Software; SYRCLE, SYstematic Review Center for Laboratory animal Experimentation.

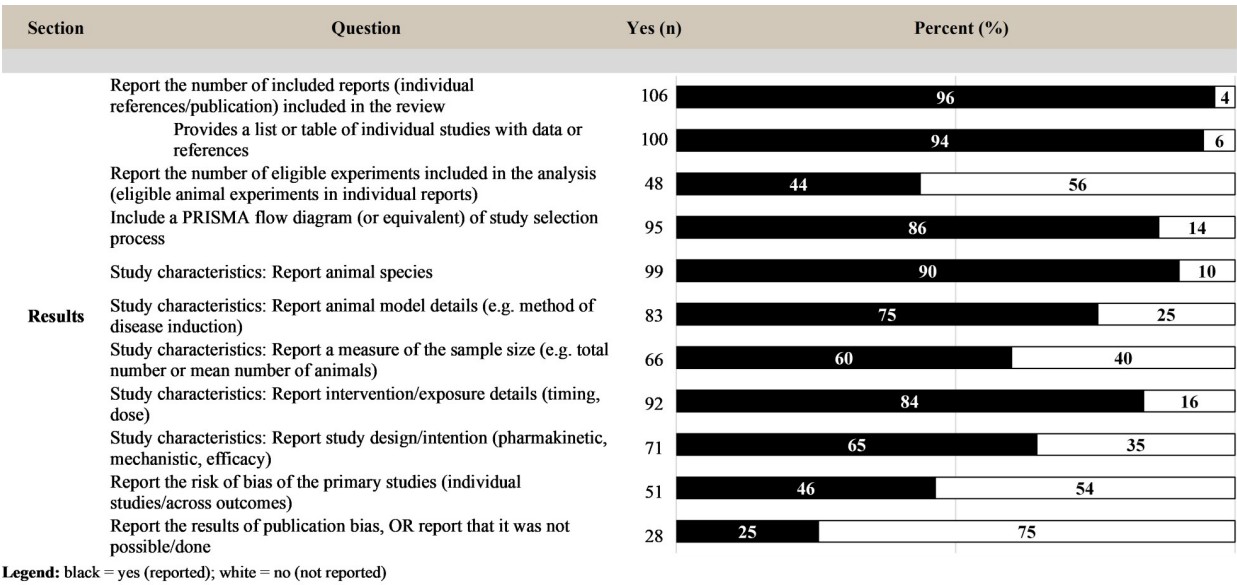

**Fig 7. Level of reporting (count and percent) for the items within the results section.** PRISMA, Preferred Reporting Items for Systematic reviews and Meta-Analyses.

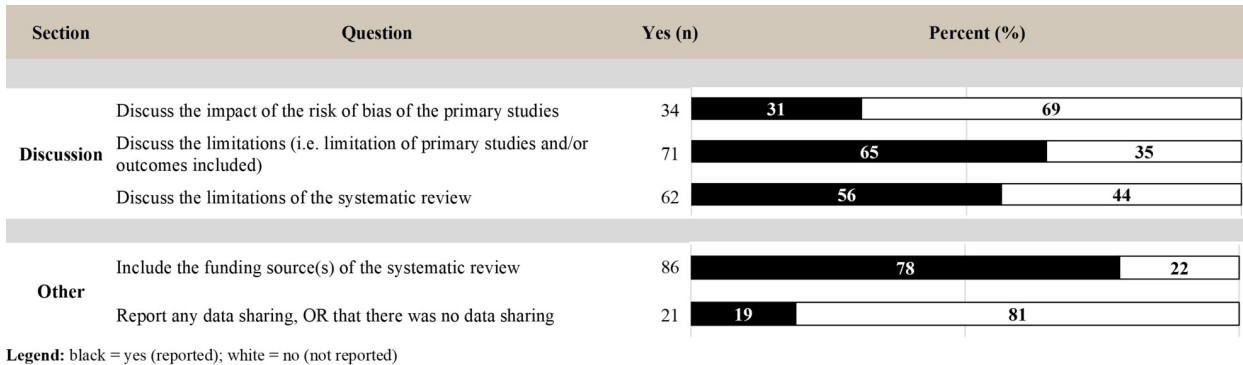

| Section | Question | Yes (n) | Percent (%) | |
|---------|----------|---------|-------------|-----|
| **Discussion** | Discuss the impact of the risk of bias of the primary studies | 34 | 31 | 69 |
| | Discuss the limitations (i.e. limitation of primary studies and/or outcomes included) | 71 | 65 | 35 |
| | Discuss the limitations of the systematic review | 62 | 56 | 44 |
| **Other** | Include the funding source(s) of the systematic review | 86 | 78 | 22 |
| | Report any data sharing, OR that there was no data sharing | 21 | 19 | 81 |

**Legend:** black = yes (reported); white = no (not reported)

**Fig 8. Level of reporting (count and percent) for the items within the discussion and other sections.**

### Method and results—For reviews with quantitative analysis

Of the 110 reviews, 44 (40%) performed a quantitative analysis. The 44 quantitative reviews investigated 17 of the 23 diseases domains: cardiovascular system disorders (9 reviews; 20%), followed by musculoskeletal and connective tissue disorders, and nervous system disorders (7; 16% each). Twenty-five (60%) quantitative reviews investigated pharmacological interventions, and 19 (40%) investigated non-pharmacological interventions. Characteristics of the 44 quantitative reviews are found in S4 and S5 Tables.

The following reporting items were specific for the quantitative reviews and were not applicable to reviews that did not perform a quantitative analysis. For the reviews that did not perform a quantitative analysis, the quantitative items were evaluated as "NA" (as described in the methods). Twenty-two (50% of 44) quantitative reviews described methods for extracting numerical data from primary studies (e.g., how data were extracted from graphical format, which is common in preclinical experimental studies). The majority (40; 95%) of quantitative reviews reported the methods for synthesizing the effect measure and methods for assessing statistical heterogeneity between studies. Fourteen reviews (32% of 44) reported methods for any data transformation needed to make extracted data suitable for analysis. Sixteen reviews (36% of 44) reported methods for handling shared control groups, and 13 (30% of 44) described methods for handling effect sizes over multiple time points, 2 common features in preclinical experimental studies. Of the 35 reviews that reported a subgroup or sensitivity analysis in the results, 33 reported the methods for these analyses. Within the results section, the confidence intervals of outcomes and a measure of heterogeneity were reported by 88% and 84% of quantitative reviews, respectively, while 29% of reviews in the sample reported the results of a subgroup or sensitivity analysis (Fig 9).

Of the 110 reviews, 46 (42%) explicitly mentioned following/using a reporting guideline or provided a completed reporting checklist. Forty-five of which reported using the PRISMA 2009 statement, and one used the Meta-analyses Of Observational Studies in Epidemiology (MOOSE) checklist.

### Discussion

This review provides a comprehensive characterization of preclinical systematic reviews and an evaluation of their reporting. Our results suggest that systematic review methodology is being applied to a diverse range of preclinical topics, and their production is increasing. Compared to the last assessment of preclinical systematic reviews published in 2014, the number of preclinical systematic reviews has nearly doubled in just 4 years. We identified that the

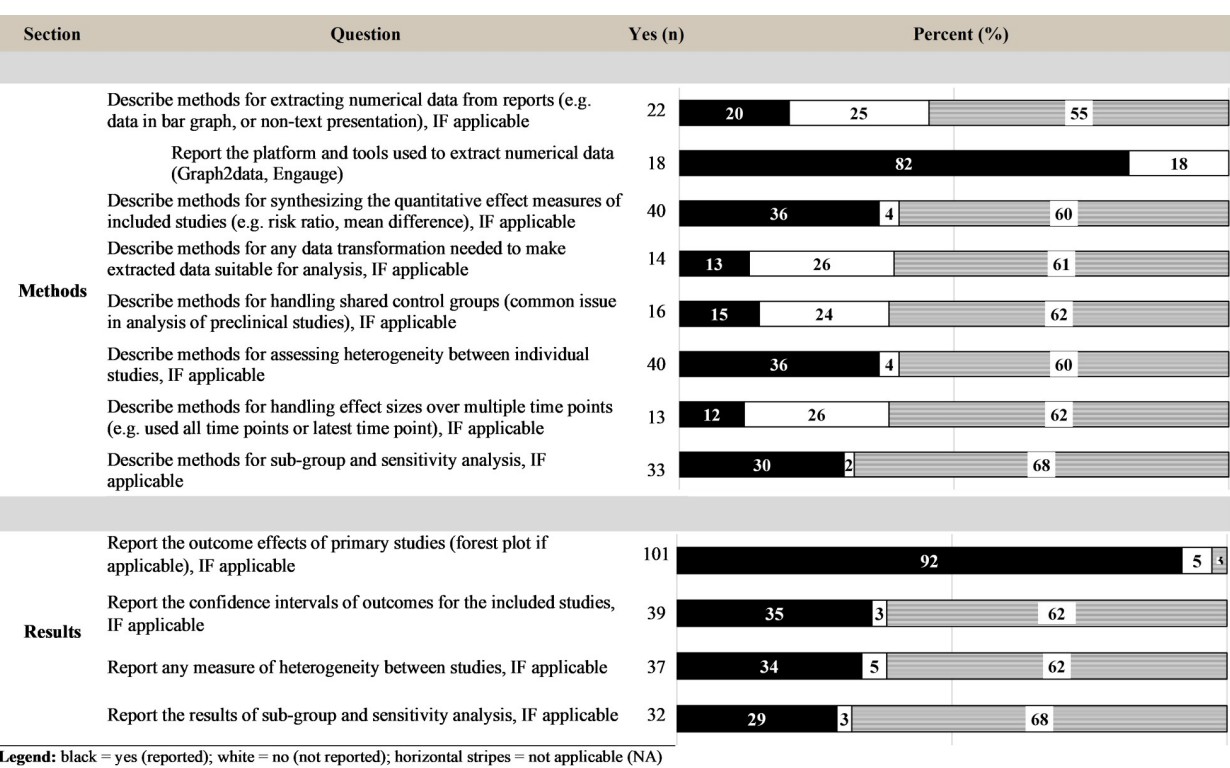

**Fig 9. Level of reporting (count and percent) for the quantitative items within the methods and results sections.**

reporting of methodology and results are not optimal. Without complete and transparent reporting of methods, it is likely not possible to gauge the trustworthiness of the results, a major limitation of any research project.

Established guidelines exist for both reporting *in vivo* preclinical experiments and clinical systematic reviews, although guidelines for reporting systematic reviews of animal experiments have yet to be developed. Similar to a landmark evaluation of clinical systematic reviews prior to the development of the 2009 PRISMA guidelines [16] and for the PRISMA 2020 update [17], we created a checklist of reporting items to assess, and we chose a sample of articles that could be feasibly evaluated. Many of the included reviews reported that they followed systematic review reporting guidelines (e.g., the PRISMA statement); however, most did not report on all required items. Fundamental items rarely reported included providing a search strategy, reporting a review protocol, describing methods for assessing the risk of bias, reporting the limitations of the review, and data sharing. Our findings are similar to an evaluation of 300 clinical systematic reviews indexed in February 2014, which found that at least a third of the reviews did not report use of a review protocol, years of coverage of the search, methods for data extraction, and methods for study risk of bias assessment [18].

Encouragingly, some of the reporting items assessed in previous reporting assessments have improved in our evaluation. Mueller and colleagues [5] found that 62% of the reviews they identified provided a flow diagram of included studies; we found that 86% did so (Z-test, $p < 0.0001$). Similarly, reporting of funding also improved (78% versus 38%; Z-test, $p < 0.00001$). Fourteen percent of reviews assessed publication bias in the previous reporting assessment, while 25% of reviews reported the results of publication bias in our updated evaluation (Z-test, $p = 0.01$). We speculate that the improved reporting of some items may be

explained in part by the increased funding provided to more recent reviews as well as established leadership by groups such as CAMARADES and SYRCLE. Since its establishment in 2004, CAMARADES has published several resources to guide investigators in the conduct and reporting of preclinical systematic reviews [3,12,19]. Additional to these resources, SYRCLE disseminated tools specifically for writing a protocol for a preclinical systematic review [20], searching for animal studies [21–23], and assessing risk of bias of preclinical studies in 2014 [24]. Also, the international registry for systematic review protocols (PROSPERO, https://www.crd.york.ac.uk/PROSPERO/) began accepting protocols for systematic reviews of animal studies relevant to human health in October 2017, which may have further improved their planning and conduct.

The improvements we see in our review provide evidence that these initiatives may have contributed to better reporting quality in preclinical systematic reviews; however, significant deficits still exist with half of studies failing to assess risk of bias. In addition, items that are unique to preclinical systematic reviews (e.g., construct validity [15]) were not evaluated objectively. Although authors may speculate on translational potential in the discussion section of papers, it remains rare for reproducible frameworks to be used to evaluate it. In part, this may reflect a lack of consensus on the definition of "construct validity"; although the concept of translation is often discussed, variable terms and approaches have been suggested (e.g., translation as a component of external validity [25], using a unique term such as "translational validity" [26], etc.). Defining this concept further through future research may be warranted. Overall, identified issues with reporting suggest that further guidance, such as a PRISMA extension specific for preclinical systematic reviews, may further improve reporting practices. Indeed, previous research has demonstrated that journals endorsing the PRISMA guidelines publish systematic reviews that are more completely reported and of higher quality [27,28]. These guidelines will need to be supported by further education and incentives in order to ensure good uptake and appropriate application, such as journals requiring the use and inclusion of the reporting checklist within their published articles.

In addition to the development of reporting guidelines, other initiatives must be considered to improve the state of reporting and quality of preclinical systematics reviews. Journals, funders, and reviewers could contribute to this improvement by advocating and appealing for data sharing, registration of protocols *a priori*, and recommending the use of tools and resources created by SYRCLE, CAMARADES, and similar groups. Moreover, reporting transparency could be improved by moving beyond simple endorsement and instead enforcing the use of a reporting checklist.

### Strengths and limitations

A strength of this study is our use of a sensitive search strategy to identify systematic reviews of *in vivo* animal experiments. However, the scope of our review may be seen as a limitation as our search focused upon *in vivo* research of largely therapeutic interventions. It should be acknowledged that *in vitro* and *ex vivo* are important areas of preclinical research, and thus the results from this study may not be representative of all preclinical systematic reviews. Furthermore, our sample was restricted to systematic reviews published in the English language, which may have caused the omission of relevant data and the introduction of bias.

A potential limitation is the timeframe of our sample, as we included reviews from 2015 to 2018 inclusively. Although we chose this sample to capture the state of reporting in preclinical systematic reviews after the previous assessment in 2014, we acknowledge that the state of reporting may have changed from 2018; however, it is important to note that no major initiatives to address systematic review reporting have occurred since that time.

Additionally, to assess the state of reporting, we selected a sample of 25% of the identified reviews. Moreover, we applied eligibility criteria for inclusion in the reporting assessment with the aim of ensuring the reviews were predominantly preclinical *in vivo* animal reviews, rather than clinical reviews with a small amount of animal data included. Both of these may create bias, as the sample may not be completely representative of the full population of reviews.

In addition, some of the items in our reporting assessment may not be generalizable to other forms of preclinical systematic reviews (e.g., those not focused on therapeutic interventions).

## Future directions

Our results show that the number of preclinical systematic reviews continues to increase compared to a previous review published in 2014. Although reporting quality has demonstrated some potential improvements, there still remains room for significant improvement. This echoes the past state of reporting within clinical systematic reviews, where historic poor reporting hampered their quality and potentially limited their utility. To address the insufficient reporting and improve transparency in clinical reviews, the PRISMA statement was developed. Although observational studies have suggested that the adoption of PRISMA has led to improved reporting of systematic reviews, it does not completely facilitate reporting within reviews of preclinical animal experiments. This is likely due to the unique differences between clinical and animal research. Specifically, our results provide rationale for a preclinical animal experiments extension of PRISMA and highlight areas of most deficient reporting. This will inform the development of a PRISMA preclinical systematic review extension.

## Supporting information

**S1 Checklist. Reporting checklist used in the reporting assessment.**
(DOCX)

**S1 Appendix. Systematic search strategy.**
(DOCX)

**S1 Table. Countries that have published a preclinical systematic review.**
(DOCX)

**S2 Table. Animal species reported within all preclinical systematic reviews.**
(DOCX)

**S3 Table. Reporting assessments for sample of 110 preclinical systematic reviews.**
(XLSX)

**S4 Table. Disease domains investigated in the preclinical systematic reviews in subgroup of studies performing quantitative analyses.**
(DOCX)

**S5 Table. Intervention and intervention subgroups evaluated in the preclinical systematic reviews in subgroup of studies performing quantitative analyses.**
(DOCX)

## Acknowledgments

We thank Drs. Olavo Amaral, Carlijn Hooijmans, Bin Ma, and Daniele Wikoff for their co-leadership on the larger research program to generate a PRISMA Extension for Preclinical *In Vivo* Animal Experiments.

## Author Contributions

**Conceptualization:** Marc T. Avey, Kimberley E. Wever, Manoj M. Lalu.

**Formal analysis:** Victoria T. Hunniford, Joshua Montroy.

**Investigation:** Victoria T. Hunniford, Joshua Montroy, Marc T. Avey, Kimberley E. Wever, Madison Foster, Grace Fox, Mackenzie Lafreniere, Mira Ghaly, Sydney Mannell, Karolina Godwinska, Avonae Gentles, Shehab Selim, Jenna MacNeil, Manoj M. Lalu.

**Methodology:** Victoria T. Hunniford, Joshua Montroy, Sarah K. McCann, Lindsey Sikora, Emily S. Sena, Matthew J. Page, Malcolm Macleod, David Moher, Manoj M. Lalu.

**Resources:** Dean A. Fergusson, Manoj M. Lalu.

**Supervision:** Dean A. Fergusson, Manoj M. Lalu.

**Writing – original draft:** Victoria T. Hunniford, Manoj M. Lalu.

**Writing – review & editing:** Victoria T. Hunniford, Joshua Montroy, Dean A. Fergusson, Marc T. Avey, Kimberley E. Wever, Sarah K. McCann, Madison Foster, Grace Fox, Mackenzie Lafreniere, Mira Ghaly, Sydney Mannell, Karolina Godwinska, Avonae Gentles, Shehab Selim, Jenna MacNeil, Lindsey Sikora, Emily S. Sena, Matthew J. Page, Malcolm Macleod, David Moher, Manoj M. Lalu.

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
