## [Editor Report · Decision Letter 0]

26 Oct 2020

Dear Dr Lalu, 

Thank you for submitting your manuscript entitled "Epidemiology and Reporting Characteristics of Preclinical Systematic Reviews" for consideration as a Meta-Research Article by PLOS Biology.

Your manuscript has now been evaluated by the PLOS Biology editorial staff, as well as by an academic editor with relevant expertise, and I'm writing to let you know that we would like to send your submission out for external peer review. Please accept my apologies for the delay in providing you with an initial decision.

Please re-submit your manuscript within two working days, i.e. by Oct 28 2020 11:59PM.

Kind regards,

Roli Roberts

Senior Editor

PLOS Biology

---

## [Decision Letter · Decision Letter 1]

11 Dec 2020

Dear Dr Lalu,

Thank you very much for submitting your manuscript " Epidemiology and Reporting Characteristics of Preclinical Systematic Reviews " for consideration as a Research Article at PLOS Biology. Your manuscript has been evaluated by the PLOS Biology editors, an Academic Editor with relevant expertise, and by three several independent reviewers.

You'll see that the reviewers are broadly positive about your study, but between them they raise a number of concerns about possible coding artefacts, the need to assess trends over time, more discussion of remedies, clarifications of the motivation and some choices made, and the need for an update to the data (we note that the search date is nearly two years ago). These issues will need to be addressed for further consideration.

In light of the reviews (below), we will not be able to accept the current version of the manuscript, but we would welcome re-submission of a much-revised version that takes into account the reviewers' comments. We cannot make any decision about publication until we have seen the revised manuscript and your response to the reviewers' comments. Your revised manuscript is also likely to be sent for further evaluation by the reviewers.

We expect to receive your revised manuscript within 3 months. 

**IMPORTANT - SUBMITTING YOUR REVISION**

*Re-submission Checklist*

*Published Peer Review*

*PLOS Data Policy*

*Blot and Gel Data Policy*

Sincerely,

Roli Roberts

Senior Editor,

rroberts@plos.org,

PLOS Biology

REVIEWERS' COMMENTS:

Reviewer #1:

[identifies himself as Kieron Rooney]

Edits:

Typographical:

Line 120: delete "was defined" which is repetitive

Lines 199 - 205 and figure 1. There are distinct differences in the numbers reported in the text and figure that need to be checked and corrected. For example - line 199 states 1549 papers were excluded, however figure 1 identifies 1585; line 200 states 807 papers were retrieved but the figure says 771; line 201 stated 325 papers were excluded but figure 1 states 329

Line 209 states 14% table 1 says 15%

Line 229 states 21 different disease domains; Table 2 identifies 23

Editorial:

Lines 146 - 148 and elsewhere for example line 168. When describing how the articles were screened and excluded etc it would be good to specifically identify which authors participated in the various tasks rather than just say "two authors …" or "we".

Line 236 and 237: the use of the word "had" seems so lax and "reported" would be more accurate representation.

Lines 238 - 241 represent the relative % of occurrence to the 203 non-pharmacological sub-set yet table 3 is relative to the total 442 excluded. I don't really care which is used, but only one reference point should be used for consistency.

Figure 4: I think the relative number of the final 110 that are pharmacological and non-pharmacological should be identified.

Line 337: the sentence that ends at "increase" on line 376 is broken by the inclusion of "compared to a previous review published in 2014". I think this first half of line 337 could be deleted.

Specific comments requiring response:

Lines 266-267 present a result in a different manner / context in which the checklist was implemented and this raises a concern regarding the process in which multiple authors were stated as implementing the assessment with consensus or third review. For example, we are given a result here of "eligible intervention timing (prevention vs rescue)" Yet in the checklist the qualifier "prevention vs rescue" is in regard to "intended goal of the intervention". I am concerned then that assessors of the articles may have scored this item differently as a result of inconsistent procedure. Can the authors please confirm if item 12 was assessed consistently between authors AND was the "prevention vs rescue" element assessed regarding whether it appeared within the context of intervention timing or intended goal of the intervention. Finally, can the authors also confirm how the potential for such a discrepancy was confirmed prior to submission.

Checklist item 6 states to assess only whether or not a protocol was registered a priori however the results on lines 261-262 identify that it was also identified if a database was indicated. In this instance then item 6 should be updated as identifying 1) was a protocol registered and 2) is the database or ID provided so as to better represent how it must have been implemented by the authors here.

Checklist item 20 - construct validity is identified as being described 0% in all studies. However, I wonder if this is simply an artefact of the checklist placing this item in the methods section. As an author of pre-clinical SRs reporting on interventions related to human clinical conditions, this is an item my colleagues and I have referred to in the discussion in the absence of a strong tool for assessing construct validity. The authors of this manuscript identify in the discussion that this concept is an emerging area of assessment. I wonder if the result presented here was re-assessed with a critical eye over attempts by authors of the SR to comment on construct validity in the discussion would identify a different outcome to be reported in this manuscript? Could the authors comment on why that item was placed only in the methods, and whether or not - even if the SRs reviewed included comment on construct validity in discussion - they scored 0 and whether or not reviewing the discussion sections of the included articles now and reporting on item 20 as an alternate discussion item could be a worthwhile adventure.

Lines 298-312 and figure 5E identify that the checklist items presented as supplementary 4 are a conflated list of two checklist - there are some questions apparently only relevant to SRs with quantitative analysis. This should have been more explicitly articulated in the method section and then also in the checklist items that are only relevant to some SRs but not all should be clearly identified. If this work is (as the authors seem to hope) to inform a PRISMA extension, then it is vital that any potentially "irrelevant" questions are clearly articulated.

Lines 298-312 sub-group analysis: I would like to at least know how many of the 42 reviews now included in a sub-analysis implemented a pharmacological or non-pharmacological intervention. The potential need for a re-analysis / updated presentation of the results in this section then may be needed or at least commented on if the authors suspect their results here may have been impacted by the distribution across those interventions. Further, an identification of the disease domains for the 110 and 42 should be provided so one can tell if the results presented may have been impacted by a non-representative sub-group relative to the original 442.

Checklist item ?? - I do not have a number to cross reference to as this item does not yet exist, but I'd have appreciated a specific question that asks whether or not the quantitative analysis in SRs grouped species or performed sub-group analyses as the contribution of species differences can often be ignored in analyses of heterogeneity.

Lines 364 - 375 strengths and limitations and, lines 121-122, and the general issue of including an item on construct validity. I have a concern regarding how the determination of the purpose of the SRs included was performed. It is not clear, if the authors of the original SR had to explicitly state they performed the SR for the purpose of informing construct validity / pre-clinical scope / translation to humans OR if it was the authors of this manuscript that inferred that purpose on the SR. For example, many pre-clinical SRs of interventions that would have met this inclusion criteria have been conducted NOT for the explicit purpose of informing human clinical practice, but rather to synthesise the current existing evidence to inform a future study design - best model, most appropriate species, etc. I would like to know if the authors of this manuscript specifically assessed if the originating SR authors explicitly stated an intent that the SR would enhance or inform translation to humans, or if that was inferred by the current authors. If it was simply assumed by the current authors then this should be included as a limitation of the analysis since it is possible that within the 75% of SRs not assessed there are better conducted SRs as their purpose was different.

Reviewer #2:

[identifies herself as Miranda W. Langendam]

The authors of this manuscript performed a cross sectional study of a sample of recent preclinical systematic reviews and examined characteristics of the systematic reviews, including reporting characteristics. To describe the reporting characteristics, they developed and used a 46-item checklist. The manuscript is informative and well-written, and addresses an important topic in preclinical research. 

Some aspects of the manuscript puzzle me, however, and I have some concerns.

First, from the Introduction it is unclear what the rationale is for a preclinical PRISMA extension. Please describe for what aspects preclinical systematic reviews are different from clinical systematic reviews, and why these aspects need additional reporting items? 

Second, the authors developed a 46-item reporting checklist, based on the PRISMA 2009 checklist. Please report what the additional items were? How was decided to include these items, what was the methodological process? 

Overall it is unclear if the 46-item checklist should be seen as the PRISMA extension, or how the presented checklist will be developed further, making the results in the current manuscript preliminary. The protocol suggests a large research project, how does this manuscript fit in the larger project?

Third, did the authors analyze trends over time in the characteristics, and if so, what where the results? Is there any improvement? Did they analyze the associations between the general characteristics (for example the type of intervention) and reporting? 

Fourth, the authors make a too strong conclusion. The authors conclude that although a considerable number of preclinical systematic reviews have been conducted, their quality and rigour is inconsistent. The authors describe only if some quality-related characteristics, for example risk of bias assessment, were reported. They did not assess if the correct risk of bias instrument was used and applied in an appropriate way. In other words, reporting, not methodological quality was assessed. 

Minor concerns:

Line 330: Many of the included reviews reported that they followed systematic review reporting guidelines: what guidance where they referring to? 

The search date is March 2019, please consider an update of the search.

Non-English systematic reviews are excluded. Please elaborate if this could cause bias, as we know that many non-English systematic reviews are published.

In the manuscript there is no reference to S4, the checklist.

Line 120: typo, please delete 'was defined'. 

Regarding the title: it may be me, but I find the term 'epidemiology' a bit odd when the manuscript is about characteristics of systematic reviews. 

Reviewer #3: 

[identifies himself as David Mellor]

The authors provide a useful snapshot of reporting quality of a population of preclinical systematic reviews. This information is vital to the community of researchers who are monitoring such changes over time. Below I provide some recommendations for modest improvements and hope to see this work published in the near future. 

An important improvement to this paper would be to more fully compare this to previous work (notably, Mueller et al 2014) and to evaluate items that have improved, not changed, or perhaps decreased in the past 6 years. You begin to do this in line 340-344 by comparing 3 items between these two time periods, but I think a more comprehensive comparison would be very helpful to users of this information. Ideally, a table that compares the items included by you, by Mueller, and the rates of each item in common would be helpful. I did not make a side by side comparison between the two checklists, but this would have the further benefit of making such a comparison possible and would provide the opportunity for you to explain any differences between the two. This comparison would make later suggestions and next directions more meaningful. 

The authors note four possible reasons for the relative improvement in the past 6+ years (again, I would like to see a better comparison over time to really see if there is much change): increased funding, leadership provided by CAMARADES and SYRCLE, establishment of PROSPERO, and journal endorsement of PROSPERO. I think that those are all reasonable explanations, but is there any more explanation possible? Without a good sense of the magnitude of the trend, it is hard to ascertain how much explanation is really needed or expected, but if the authors provide more comparison and find substantial change, some more explanation of the possible causes would be helpful. Even if that is not possible, the authors could acknowledge that. 

I believe the article would be improved with greater assertions on steps that authors, funders, reviewers, and journals should take to further improve the situation. Some possibilities could include journal requirements for systematic reviews to more fully report details (journal mandates are sometimes effective at changing behaviors such as data sharing https://osf.io/preprints/bitss/39cfb/ but not others such as registration https://jamanetwork.com/journals/jamainternalmedicine/fullarticle/2727849 or use of reporting checklists http://journals.plos.org/plosone/article?id=10.1371/journal.pone.0183591 ) recognition for desired behaviors, eg badges http://journals.plos.org/plosbiology/article?id=10.1371/journal.pbio.1002456 or review of systematic review protocols such as is conducted by Cochrane or Campbell reviews. 

Data availability: The authors provide their complete search strategy, checklist, and protocol. However, I was not able to easily find any raw, paper-level data. Even if the authors removed the identifying information from the file (which I don't think would be necessary, but if they want to prevent any single paper as being identified as "worst" then perhaps they could do so), such a file could be helpful. For example, correlations between excluded items could reveal meaningful patterns for future work. If the authors did provide it and I am missing it, please forgive me- although I would recommend adding it to the shared drive along with a code book or README file to explain any odd variables. 

I regularly sign my reviews. I hope that the provided comments are helpful to the authors.

Sincerely, David Mellor, Director of Policy Initiatives, Center for Open Science

Minor comments

Line 99: "The prevalence and quality of preclinical systematic reviews has not been formally evaluated since 2014." It is unclear if the authors are referring to Mueller et al 2014 https://doi.org/10.1371/journal.pone.0116016 or Page et al 2016 https://doi.org/10.1371/journal.pmed.1002028 (which reviewed the literature up to 2014), or both. My other comments focus a bit on asking the authors to comment on some trends over time, so this comparison will be necessary.

Line 106: It would probably be helpful to point directly to the protocol, https://osf.io/9mzsv/

Line 120: Typo: "We defined preclinical was defined as research…."

Line 123: Please justify the exclusion of in vitro and ex vivo studies (presumably this is to make it comparable with Mueller et al 2014, but please state. 

Line 162: "We next evaluated the quality of reporting in a random sample of 25% included systematic reviews. This sample size was chosen based on available resources." It would be helpful for other researchers to provide information on time required to score the 110 articles (either here or around line 246). Not providing this information would be acceptable (perhaps time cannot be reasonably estimated if it was not tracked during the screening), as it is a bit beyond the scope of normal reporting expectations, but would nonetheless be a useful piece of information for others. 

Line 225, Figure 3: It's unclear why this information is provided in a pyramid shape. The shape suggests more meaning than I think is intended by the authors. Of course, rat is the most common animal, thus at the top of the pyramid, but is not necessarily in a different class or tier than mouse, with nearly as many animals. I think the visual of the animal outlines is a nice way to summarize the information but rearranging into a circle or square would remove any suggestion that the levels are meaningful (beyond simple frequency).

Line 261: "Twenty-two reviews (20%) reported a protocol had been developed a priori…"

---

## [Editor Report · Decision Letter 2]

5 Mar 2021

Dear Dr Lalu,

On behalf of my colleagues and the Academic Editor, Lisa Bero, I'm pleased to say that we can in principle offer to publish your Meta-Research Article "Epidemiology and Reporting Characteristics of Preclinical Systematic Reviews" in PLOS Biology, provided you address any remaining formatting and reporting issues. These will be detailed in an email that will follow this letter and that you will usually receive within 2-3 business days, during which time no action is required from you. Please note that we will not be able to formally accept your manuscript and schedule it for publication until you have made the required changes.

PRESS: We frequently collaborate with press offices. If your institution or institutions have a press office, please notify them about your upcoming paper at this point, to enable them to help maximise its impact. If the press office is planning to promote your findings, we would be grateful if they could coordinate with biologypress@plos.org. If you have not yet opted out of the early version process, we ask that you notify us immediately of any press plans so that we may do so on your behalf.

Thank you again for supporting Open Access publishing. We look forward to publishing your paper in PLOS Biology. 

Sincerely,

Roli Roberts

Roland G Roberts, PhD 

Senior Editor 

PLOS Biology